# Pre-Pregnancy Weight and Symptoms of Attention Deficit Hyperactivity Disorder and Executive Functioning Behaviors in Preschool Children

**DOI:** 10.3390/ijerph16040667

**Published:** 2019-02-25

**Authors:** Bernard F. Fuemmeler, Nancy Zucker, Yaou Sheng, Carmen E. Sanchez, Rachel Maguire, Susan K. Murphy, Scott H. Kollins, Cathrine Hoyo

**Affiliations:** 1Department of Health Behavior and Policy, Virginia Commonwealth University, Richmond, VA 23219, USA; yaou.sheng@vcuhealth.org; 2Department of Psychiatry and Behavioral Sciences, Duke University Medical Center, Durham, NC 27705, USA; nancy.zucker@duke.edu (N.Z.); Carmen.Sanchez@neurocogtrials.com (C.E.Z.); susan.murphy@duke.edu (S.K.M.); scott.kollins@duke.edu (S.H.K.); 3Department of Biological Sciences, North Carolina State University, Raleigh, NC 27695, USA; rlmaguir@ncsu.edu (R.M.); choyo@ncsu.edu (C.H.)

**Keywords:** pregnancy, gestational weight gain, maternal obesity, child development, ADHD

## Abstract

This study examines pre-pregnancy Body Mass Index (BMI) and gestational weight gain (GWG) in relation to early childhood Attention Deficit Hyperactivity Disorder (ADHD) symptoms and related executive self-regulation behaviors. The analyses sample (*n* = 331) included a subsample of participants from a birth cohort recruited from prenatal clinics and hospital facilities from April 2005 to June 2011 in Durham, North Carolina. Pre-pregnancy BMI was calculated from weight at the last menstrual period and height was extracted from medical records. Gestational weight gain was calculated from pre-pregnancy weight and weight measured at the time of delivery. ADHD symptoms and executive self-regulation behaviors were assessed by maternal report (mean age = 3 years). Multivariable regression methods with inverse probability weighting (IPW) were used to evaluate associations accounting for sample selection bias and confounding. Pre-pregnancy BMI at levels ≥35 was positively associated with higher ADHD symptoms and worse executive self-regulation behaviors (inhibitory control and attention). Compared to adequate GWG, less than adequate GWG was related to more ADHD hyperactive-impulsive symptoms, whereas greater than adequate GWG was related to more problematic behaviors related to working memory and planning. The findings support a link between maternal weight and child neurodevelopment. Continued research that help identify biological mechanisms are needed.

## 1. Introduction

Globally, more than 10% of the population have a body mass index (BMI) in the obese range (BMI ≥ 30) with the prevalence having doubled in 70 countries in the past 25 years [1]. In the United States, the age-adjusted prevalence is higher among women compared to men (40.1% vs. 35%) [2]. And either at conception or during their pregnancy, 1 in 4 women have a BMI in the obese range [3]. Maternal obesity during pregnancy has critical health consequences for the mother, as well as acute and enduring metabolic consequences on the developing child [4].

Recent studies further suggest that maternal obesity prior to or during pregnancy may also be related to cognitive and neurodevelopmental problems in offspring. For instance, animal studies indicate that an obese state during pregnancy leads to a cascading series of events that affect brain development and subsequent neurodevelopmental functioning. Bilbo and Tsang (2010) showed in rats that gestational obesity produces changes in inflammatory signaling that has downstream effects on fetal microglial functioning, brain development and subsequent offspring spatial learning abilities [5]. In addition, obesity during pregnancy may also affect brain development via metabolic hormone-induced changes, such as exposure to increased circulating levels of leptin or insulin, or suppression of serotoninergic functions that disrupt the normal development of the serotonin (5HT) system [6]. It is also speculated that maternal dietary quality (e.g., high fat diet) may induce changes in fetal dopamine signaling and reward-related neural system development with consequences for offspring executive functions [7].

Studies of humans align with these findings from studies of animals. Maternal pre-pregnancy obesity has been linked with a number of cognitive and neurodevelopmental outcomes in offspring, such as cognitive/intellectual delay [8], poorer academic achievement scores [9], behavioral/emotional regulation problems [10] and an increased risk of Attention Deficit Hyperactivity Disorder (ADHD), Autism Spectrum Disorder (ASD) symptoms [11], and other developmental delays (DD) [12,13,14]. A recent meta-analysis of 32 studies found that relative to children born to normal weight mothers, children born to mothers who were obese during pregnancy were at greater risk for having a compromised cognitive or neurodevelopment outcome (OR = 1.51). Within this body of literature, ADHD and related symptoms was the least investigated outcome (*n* = 7 in meta-analysis), but had the highest odds ratio (1.62) associated with pre-pregnancy obesity [15]. Of note, however, studies have also documented null associations between pre-pregnancy weight and neurodevelopmental outcomes [16,17], and discordant sibling design studies suggested that the association may not be causal [18,19]. Given some of the differences in findings, continued research is needed using different samples and different methods to better understand this association between pre-pregnancy obesity and child neurodevelopmental outcomes.

Understanding prenatal and developmental factors associated with ADHD and poor executive self-regulation is important since childhood symptoms in these domains, independent of a clinical diagnosis, can increase the risk for a number of unhealthy behaviors later in adulthood [20,21] and has a large economic burden [22,23,24]. Thus, to better inform antenatal prevention and postnatal risk-identification/remediation strategies, continued research is needed to further clarify early risk factors for ADHD symptoms and poor executive self-regulation, including the extent to which pre-pregnancy obesity is related to childhood ADHD symptoms and related behaviors.

The purpose of this study was to add additional data to this growing body of literature. First, this study includes assessment of both ADHD symptoms and behaviors in everyday life that depend on executive self-regulation processes (e.g., those that involve negotiation of competing desired outcomes). Second, whereas existing studies have focused primarily on school age children, the study population herein is a sample of preschool age children. If associations are present at the preschool age, it may provide further justification for earlier monitoring of behavioral problems among children born to mothers obese prior to and during pregnancy. Third, we examined both continuous and categorical indicators of pre-pregnancy BMI and gestational weight gain (GWG) in order to identify associations at varying levels of exposure. In addition, we used inverse probability weighting (IPW), a robust method for accounting for sample selection bias and confounding in observational studies [25,26]. In light of existing research, we hypothesize that children born to mothers with higher pre-pregnancy BMIs or greater GWG will display greater ADHD symptoms and poorer executive self-regulation behaviors.

## 2. Methods

### 2.1. Study Sample

Participants were enrolled as part of the Newborn Epigenetics STudy (NEST), a birth cohort designed to identify the effects of early exposures on epigenetic profiles and developmental outcomes. Briefly, 2595 pregnant women were recruited from prenatal clinics serving Duke University Hospital and Durham Regional Hospital Obstetric facilities in Durham, North Carolina, from April 2005 to June 2011. Eligibility criteria was: Age ≥ 18 years, English speaking, pregnant, and intention to use one of the two obstetrics facilities to enable access to labor and birth outcome data. The Institutional Review Board approved the study procedures.

Between January 2011 and June 2013, mothers from the NEST study, who had children between the ages of 2 and 6 years of age (*n* = 1278) were sent a survey by research staff, that included measures of ADHD symptoms and related executive behaviors. Of those who were sent a survey, 223 were undeliverable and 127 were deemed ineligible due to the child’s age at the time the survey was completed. We received 531 surveys (531/(1278 − 127) = 46.4% response rate). Of the 531 returned surveys, 331 had data on all data needed for this study. Main sources of missing data were due to one measure (maternal ADHD symptoms) that was added subsequent to sending and initial set of surveys and missing data on pre-pregnancy weight and data on gestational diabetes from the medical records. The mean gestational age at enrollment for the analysis cohort was 19 weeks (range: 5–42 weeks).

We compared our analysis sample (*n* = 331) to the sample who were originally sent surveys minus those who returned the survey (*n* = 331) and those who returned from mothers who completed the survey after their child turned 6 years of age (*n* = 127). Compared to eligible mothers who did not return the sent survey, the analytic sample was slightly older (age = 29 years vs. 28 years, *p* = 0.01), with more college graduates (51% vs. 28%, *χ*^2^ = 57.38, *p* < 0.001), significantly more Caucasians (46% vs. 30%, *χ*^2^ = 24.83, *p* < 0.001), and no differences for African Americans (47% vs. 52%, *χ*^2^ = 2.22, *p* = 0.14), or Hispanics (3% vs. 2%, *χ*^2^ = 0.39, *p* = 0.53). Fewer mothers reported “Yes” for prenatal smoking status in analysis sample (21% vs. 31%, *χ*^2^ = 10.76, *p* = 0.001), which were documented as binary variables. The two samples were similar with respect to infant sex ((male) 51.7% vs. 50.9%, *χ*^2^ = 0.06, *p* = 0.81). Mean pre-pregnancy BMI was similar (28.12 vs. 28.85, *p* = 0.21), however, mean GWG was greater in the analysis sample (15.82 vs. 13.86, *p* = 0.001). Proportions of women having gestational diabetes were similar in the two samples (7% vs. 8%, *χ*^2^ = 0.23, *p* = 0.63). The propensity scoring method described below was used to account for non-response selection bias in our sample.

### 2.2. Measures

#### 2.2.1. Maternal Anthropometrics

All prenatal clinics used the same model of Stadiometer and weight scale (Seca Corporation, Columbia, MD, USA) with height being accurate within 1/8 of 1 inch and weight accurate within 0.1 pounds. Pre-pregnancy Body Mass Index (BMI) was computed (kg/m^2^) from self-reported weight at the last menstrual period prior to the first prenatal visit, and nurse-measured height. World Health Organization (WHO) categories were also constructed (Underweight = 0–18.5 kg/m^2^; Normal = 18.5–24.9 kg/m^2^; Overweight = 25–29.9 kg/m^2^; Obese Class I = 30–34.9 kg/m^2^, and Obese Class II ≥ 35 kg/m^2^). To evaluate the potential recall bias in the sample, we abstracted weight from medical records of 243 women who visited a hospital affiliated clinic on average 6 months prior to their first prenatal visit for the pregnancy that made them eligible for the study. The correlation between periconceptional self-reported and nurse-measured weight was 0.98 (*p* < 0.0001). When considering only overweight or obese women, the correlation remained significant (*r* = 0.96, *p* < 0.0001).

Gestational weight gain was computed as the difference between the weight of the last menstrual period (prior to the first prenatal visit), and the last weight taken up to seven days before or at delivery from medical records [27]. Categories for gestational weight gain were constructed based on the Institute of Medicine (IOM) guidelines of 12.5–18 kg weight gain for women with pre-pregnancy weight of < 19.8 kg/m2; 11.5–16 kg weight gain for women 19.8-26.0 BMI; 7–11.5 kg/m^2^ for women 26–29 kg/m^2^ and 5–9 kg for women ≥ 30 kg/m^2^ [28]. These ranges are considered “recommended” with weight gain below the range categorized as “less than recommended” and weight gain above the range categorized as “more than recommended.”

#### 2.2.2. Child ADHD Symptoms

The Behavior Assessment System for Children (BASC), 2nd edition preschool parent rating scale was used to assess ADHD related symptoms. The BASC is a valid, reliable, 134-item measure of child behavior problems and is appropriate for use in children ages 2 through 5 years 11 months [29]. The measure has been shown to discriminate well between children with ADHD and healthy controls [30]. Raw scores for hyperactivity (HY) and attention problems (AP) were computed. Raw scores are more precise and uniform than age and sex adjusted T-scores and are generally recommended for correlative and longitudinal analyses [31]. T-scores were computed and presented for descriptive purposes. Higher HY, AP and ADHD scores suggest more problematic behaviors. Internal reliability in this sample was 0.80 for the HY subscale and 0.77 for the AP subscale.

#### 2.2.3. Child Executive Behaviors

The Behavior Rating Inventory of Executive Function (BRIEF)—Preschool Version was used to assess executive behaviors [32]. The BRIEF is a standardized rating scale with 63 items designed to measure a range of behaviors related to executive function in preschool children 2 through 5 years 11 months. Items relate to 5 primary subscales: (1) Inhibit (IN) relates to the ability to control one’s behavior; (2) shift (SF) measures the capacity to transition attention between activities; (3) emotional control (EC) assesses the ability to modulate emotions; (4) working memory (WM) taps the ability to hold information in mind; and (5) plan/organize (PO) reflects the ability to plan for future events. Raw scores for each of the lower order domains were computed along with a global executive composite (GEC) which is the sum total raw score across the five domains. Similar to the BASC, age and sex adjusted T-scores are available. As with the BRIEF, we present T-scores for descriptive purposes and used the raw scores for analysis. Higher scores reflect greater problems across these domains. Internal reliability coefficients in this sample ranged from 0.83–0.91 for the 5 subscales.

#### 2.2.4. Covariates and Other Variables

The Adult ADHD Self-Report Scale (ASRS) screener was used to assess ADHD symptoms among the mothers [33]. The ASRS screener consists of 6 items from the full 18 item version which is based on the Diagnostic and Statistical Manual—IV symptoms of ADHD. The 6-item screener has been shown to have high sensitivity (68.7%), specificity (99.5%) and total classification accuracy (97.9%) in a US population based cohort [33].

At enrollment (mean gestation age 19 weeks), a self- or interviewer-administered questionnaire solicited information about maternal age, race, educational status, parity, and smoking history (whether they smoked prior to or during pregnancy). Birth weight, gestational weeks, and infant sex were obtained from medical records and child age was calculated from their birth date and the date the mother completed the follow-up survey.

### 2.3. Statistical Analyses

To examine the associations between pre-pregnancy BMI and the two outcomes (ADHD symptoms and executive behaviors), multivariable linear regression with inverse probability weighting (IPW) was used to regress pre-pregnancy BMI as a continuous variable (Model 1) and as a categorical variable using the WHO groups ((underweight/normal, overweight, obese class I, and obese class II/III (Model 2)) on BASC ADHD symptoms and BRIEF scores. IPW is a robust method for addressing confounding in observational studies by simulating a randomized design such that the distribution of confounding variables are balanced across the exposure [26]. Variables selected for the models were associated with the BASC ADHD symptoms or any of the BRIEF scores at *p* < 0.1. These variables included gestational diabetes, parity, birth weight, the child’s age, sex, mother’s age, race, education, gestational weeks, smoking and mother’s concurrent ADHD. The IPW method used addressed bias from two sources: nonresponse selection and confounding. Following the methods described by Rinsky et al. (2017) [26] logistic regression models were performed to estimate (1) stabilized inverse probability of response weights (IPRWs) and (2) stabilized inverse probability of exposure weights (IPEWs). The final weight was as the product of the IPRWs and IPEWs, which were applied to the regression models [25,26]. All regression models were fit using robust standard errors. Similar models were used for GWG as a continuous variable (Model 3) and as a categorical variable defined by the Institute of Medicine (IOM) recommended cut points (Model 4) and additionally adjusting for pre-pregnancy BMI. By design, we excluded children who were born moderately to extremely preterm (<34 gestational weeks) as existing evidence indicates that children born preterm are at greater risk for ADHD symptoms [34]. We did run models that included all cases regardless of gestational age and excluded control for birth weight since these might be considered intermediating variables; the pattern of results were similar to those found below. Additionally, we conducted a sensitivity analysis, using a more conservative inclusion criteria by limiting analyses to children born ≥ 37 weeks. STATA 14.1 (StataCorp LLC, College Station, TX, USA) and SAS 9.4 statistical packages (SAS Institute Inc., Cary, NC, USA) were used for all analyses.

## 3. Results

Table 1 Summarizes the sample characteristics of study participants by composite BASC ADHD and BRIEF GEC scores. On average, mothers were 29 years of age (SD = 5.6) at the time of delivery, with 47% self-identifying as African American, 46% as Caucasian, 3% as Hispanic, and 5% other. Prevalence of pre-pregnancy obesity was 32%, with 60% of women gaining more than the recommended amount of weight during pregnancy. The mean age of children when mothers completed the survey was 38.1 months (SD = 14.1). Standardized scoring indicated that *n* = 36 (11%) and *n* = 54 (16%) participants had T-scores of 60 or above (“clinically significant range”) for the BASC HY and BASC AP, respectively. For BRIEF GEC, *n* = 52 (16%) participants had T-scores of 65 or above (“abnormal elevation”). Compared to the sample used to derive population norms, the percentage in the clinical significant range in our study was fairly similar to the norm sample. Specifically, the norm sample used for the BASC-II showed that among children ages 2–6 years, 14–15% for the BASC HY and 14–17% for the BASC AP, were in the clinical range. Meanwhile, norm sample from the BRIEF population showed 8–10% were in the abnormal elevation range for boys and girls at the same age group. Bivariate analyses show that several of the covariates were related to the composite BASC ADHD and BRIEF GEC scores. Further, composite BASC ADHD and BRIEF GEC scores were monotonically related to higher weight categories. Gaining less than or more than the IOM recommendations for weight gain during pregnancy were both associated with statistically higher composite BASC ADHD scores (*p* < 0.01) and marginally, but not statistically significant, with a higher BRIEF GEC score (*p* = 0.07).

### 3.1. Pre-pregnancy BMI and BASC and BRIEF Scores

As seen in Table 2, after adjusting for gestational diabetes, parity, birth weight, infant’s age, sex, mother’s age, race, education, gestational weeks, smoking and mother’s concurrent ADHD, pre-pregnancy BMI was positively associated with more total ADHD symptoms (*B* = 0.17, *p* = 0.001) and greater hyperactive-impulsive (*B* = 0.09, *p* = 0.02) and attention problems symptoms (*B* = 0.08, *p* < 0.001) (Model 1). As presented in Table 3, after adjustment for the same set of covariates, pre-pregnancy BMI was positively associated with more problems related to executive behaviors (*B* = 0.38, *p* = 0.02) (Model 1). Figure 1a and 1b display a scatter plot of the associations between pre-pregnancy BMI and the composite ADHD symptoms and BRIEF GEC scores, respectively. Pre-pregnancy BMI was associated with a more problems across a number of BRIEF subscales, including inhibit (*B* = 0.09, *p* = 0.04), shift (*B* = 0.06, *p* = 0.02), working memory (*B* = 0.12, *p* = 0.02) and planning and organizing (*B* = 0.07, *p* = 0.004).

Model 2 in Table 2 and Table 3 repeated this analytical strategy using pre-pregnancy BMI as a categorical variable. These results were similar to the pattern of results in Model 1 in Table 2 and Table 3; however, they highlight a threshold effect whereby compared to children born of mothers with a BMI in the healthy range, children born to mothers with obese class II or BMI ≥ 35 were significantly more likely to have higher ADHD symptoms and more problems related to executive behaviors.

### 3.2. Gestational Weight Gain (GWG) and BASC and BRIEF Scores

As seen in Table 2 and Table 3, treating GWG as a continuous variable, and adjusting for pre-pregnancy BMI in addition to prior covariates, was not significantly associated with ADHD symptoms and BRIEF GEC scores (Model 3). Figure 1c,d display a scatter plot of the associations between GWG and the composite ADHD and BRIEF GEC scores, respectively. When BRIEF scale scores were analyzed independently, GWG was only statistically significantly related to behaviors related to working memory (*B* = 0.08, *p* = 0.046), planning and organizing (*B* = 0.07, *p* = 0.01) and was marginally significantly related to inhibit (*B* = 2.23, *p* = 0.052).

Model 4 in Table 2 and Table 3 repeated this using IOM categories of GWG. For ADHD symptoms, the results show that compared to mothers with adequate weight gain during their pregnancy, mothers with less than adequate weight gain rated children as having higher hyperactive-impulsive symptoms (*B* = 2.04, *p* = 0.02). With respect to executive behaviors, compared to women with adequate GWG, women with greater than adequate GWG had children with elevated, but not statistically significantly higher scores related to problems inhibiting behavior (*p* < 0.10).

### 3.3. Sensitivity Analyses

Sensitivity analysis in which we limited the analyses to children born ≥ 37 weeks produced similar results to the main analyses that included children born at ≥ 34 weeks (Appendix A and Appendix A).

## 4. Discussion

Our findings demonstrate a consistent relationship between pre-pregnancy BMI ≥ 35 and child ADHD symptoms and executive self-regulation behaviors during the preschool age (e.g., difficulty shifting attention, poorer working memory, poorer planning/organization behaviors). GWG, either less than the recommended levels or more than the recommended levels, were also found to be associated with ADHD symptoms and executive self-regulation behaviors with less than recommended levels of weight gain associated with hyperactive-impulsive symptoms and more than recommended levels of weight gain associated with worse working memory, planning and organizing behavior. Of note, the results were significant using a propensity scoring method – a robust method to minimize selection bias and balance confounding across the exposure (i.e., BMI or GWG). Overall, this study contributes additional data to a growing body of research suggesting that maternal weight at conception may influence cognitive and neurodevelopmental outcomes in children and increase the likelihood of ADHD symptoms [15].

The findings are in line previously published results, especially those by Jo et al. (2015) who found that an ADHD diagnosis was significantly higher among children born to mothers with BMI ≥ 35 relative to children born to mothers with a BMI in the normal range [35]. The findings with respect to GWG controlling for pre-pregnancy BMI were a bit more nuanced in that less than adequate GWG was related to more hyperactive-impulsive symptoms and more than adequate GWG was related to more inattention symptoms. GWG is infrequently studied in this literature, however, at least one previous study found no relationship between GWG and ADHD symptoms [36]. A more recent study by Pugh et al. (2018) reported that when considering the joint contribution of pre-pregnancy BMI and GWG, children of lean mothers with low GWG and children of overweight mothers with high GWG were more likely to make commission errors (more impulsive responding) on the Connors’ Continuous Performance test [37]. It is possible that inadequate and excessive GWG may be operating through different pathways, such as nutritional deficits among women with less than adequate weight gain and excessive inflammation among those with more than adequate GWG [37]. It is unclear at this time, however, if these pathways might result in differential outcomes (inattention vs. hyperactive-impulsive problems) or if they are a shared risk factors for global problems related to cognitive and neurodevelopment problems.

The potential biological mechanisms linking these associations remain to be investigated. A prevailing explanation for these associations is that fetal exposure to disruption from inflammatory cytokines among women who have a higher BMI may alter fetal brain development [38,39]. Cytokines are implicated in brain development and are involved in synaptogenesis and refinement, apoptosis and angiogenesis [40]. There is increasing evidence of a link between inflammation and neurodevelopmental disorders, such as autism and ADHD [41,42]. Further, a recent study has also shown a significant relationship between maternal cytokine levels during gestation and executive functions related to inhibitory control and attention [43]. Continued investigation of these associations between pre-pregnancy obesity with a focus on potential biological mediating mechanisms could help to further advance research on possible mechanisms at work.

This study also adds additional data to this field by examining ADHD symptoms and executive behaviors using well-validated measures among a multiethnic sample of preschool children. The findings from this community-based cohort help support the generalization of findings from previous studies. We also examined a broader range of executive capacities; findings that may help to inform hypotheses of potential mechanisms and aid in the design of early intervention strategies. Specifically, our models revealed consistent deficits in domains of attention and cognitive functions (e.g., planning, inhibiting behavior), but not in emotional control. Pre-pregnancy BMI has been associated with emotional intensity and regulation in offspring who were on average 5 years of age [10]. Thus, it may be that at this early age, poor emotional control is more normative than hyperactive-impulsive problems and inattention, making it more difficult for mothers to assess.

### Strengths and Limitations

The results should be interpreted within the context of the study’s limitations. First, as is the case with most all studies on this topic, pre-pregnancy weight was based on maternal self-report of weight prior to her last menstrual period. There is some evidence that women who are overweight and/or obese may underreport their weight status [44]. However, we retrieved measured weight data from a medical encounter that occurred prior to the first prenatal visit on a subset of women. We found a high degree of correlation between the nurse-measured weight and periconceptional self-reported weight. The correlation remained significant among women who were overweight or had obesity, suggesting that self-reported periconceptional weight is valid in this sample even among women with BMIs in the overweight or obese range.

Second, we employed dimensional measures of ADHD symptoms and executive function behaviors, which may be related to a number of psychiatric conditions (e.g., oppositional defiant problems). As differential diagnostic evaluation was not conducted, we are not able to conclude to what extent pre-pregnancy BMI is related to ADHD, as it is clinically defined or to other forms of psychopathology. Since this sample was a community ascertained sample of young children, rather than a clinical sample, the prevalence of ADHD would likely have been low reducing our power to evaluate the hypothesized associations. Additionally, we only have parent questionnaires available for assessment of both ADHD symptoms and executive functioning. Though parents can provide information on their children’s behavior in a naturalistic setting, their report may be limited since parents may be biased in terms of how they report on their children’s behavior. Nevertheless, we did use standardized measures of ADHD and executive behaviors commonly used to assess ADHD. To overcome these measurement challenges, it would be important in future studies to use longitudinal designs with individually administered and performance-based measures (e.g., NIH toolbox, measures of IQ). Having multiple informants, such a teachers, would also be warranted in future studies. This was a challenge with this particular study as children were young and not all of them had pre-school or secondary caregivers qualified to provide reports.

Finally, it has been suggested that the associations observed between pre-pregnancy BMI and poorer cognitive and neurodevelopmental outcomes may be due to unmeasured confounding factors (e.g., socio-economic status, dietary quality, improper control of genetic/heritable traits) [18,19]. The present study does not resolve this controversy. However, strengths of the approach here are that we accounted for maternal ADHD symptoms in our analyses and used a highly robust method to more fully account for confounding factors as well as selection bias.

## 5. Conclusions

This study supports a growing body of literature both from animal and human studies that find offspring born of mothers with excess weight prior to or during pregnancy are at increased risk for cognitive and neurodevelopmental problems, including more ADHD and related executive self-regulation problems. Notably, the effect sizes observed here were small. For example, the standardized beta for pre-pregnancy BMI ≥ 35 and ADHD symptoms was 0.21. Small or medium effect sizes would be expected in studies of outcomes, such as ADHD symptoms or executive functioning behaviors, where there are likely multiple causal pathways. To put the effect we observed in perspective, the Cohen’s *d* effect size for maternal smoking during pregnancy and ADHD is ~0.48 (based on a pooled OR of 2.36 from a meta-analysis) [45]. However, the prevalence of maternal smoking during pregnancy is around 7% [46], whereas the prevalence of pre-pregnancy obesity is approximately 25%, with greater increasing trends being observed among women with more severe obesity (BMIs ≥ 35) [3]. Thus, given the current prevalence of obesity, the small effect size observed here may have an equally, if not greater, impact at the population level relative to other risk factors that are less frequent, but have a larger effect size.

Understanding factors that contribute to early childhood self-regulation problems, such as hyperactive-impulsive symptoms and problems related to attention and poor executive control is important since deficits in these areas can alter the trajectory of health and have a profound impact on their economic burden to society. For instance, Caspi, et al. (2016) [22] recently showed that variation in age-three “brain health” (a global index of early cognitive abilities and examiner rated self-regulation behaviors) significantly predicted individuals who as adults would be members of the population with a constellation of economically burdensome health outcomes (excess obesity, higher prevalence of smoking, greater hospital utilization, etc.). On an individual level, maintaining a healthy periconceptional and pregnancy weight is warranted for the overall health benefits to the woman and her fetus, but clinical action to specifically reduce ADHD in the offspring born to these women is not warranted at this time. Nevertheless, if studies continue to show similar results as shown here, periconception, prenatal and postnatal risk-identification/remediation education and communication strategies could be leveraged to help foster optimal “brain health” and lower the risk of ADHD and related executive self-regulation among pregnant women with high levels of obesity.

## Figures and Tables

**Figure 1 ijerph-16-00667-f001:**
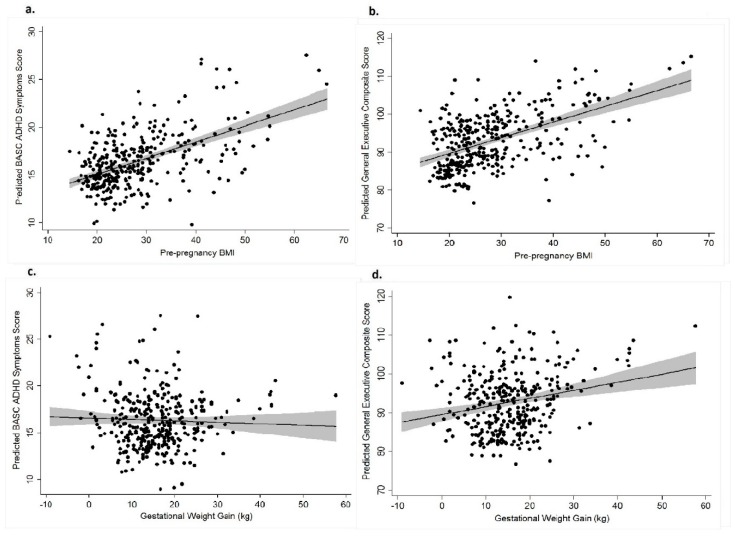
Fitted values (solid dots), fitted regression lines (solid lines) and 95% confidence limits (shaded area) for relationship between pre-pregnancy BMI and predicted BASC ADHD symptoms score (**a**), pre-pregnancy BMI and predicted BRIEF GEC score (**b**), gestational weight gain and predicted BASC ADHD symptoms score (**c**), and gestational weight gain and predicted BRIEF GEC score (**d**). Regression models adjusted for gestational diabetes, parity, birth weight, child’s age, sex, mother’s age, race, education, gestational weeks, smoking and mother’s ADHD for (**a**) and (**b**); and adjusted for pre-pregnancy BMI, gestational diabetes, parity, birth weight, child’s age, sex, mother’s age, race, education, gestational weeks, smoking and mother’s ADHD for (**c**) and (**d**).

**Table 1 ijerph-16-00667-t001:** Overall sample characteristics, BASC ADHD and BRIEF GEC by sample characteristics.

Characteristics	*n*	(%)	BASC ADHD		BRIEF GEC	
			Mean	(95% CI)	Mean	(95% CI)
Overall	331	100.0	16.32	(15.55–17.09)	92.90	(90.66–95.14)
Mother age at delivery, years						
18–24	80	24.2	17.63	(15.87–19.38)	98.23	(93.52–102.95)
25–29	91	27.5	16.68	(14.94–18.42)	92.58	(87.51–97.65)
30–34	109	32.9	15.33	(14.25–16.42)	90.05	(86.57–93.52)
35+	51	15.4	15.75	(13.98–17.51)	91.92	(86.57–97.27)
Maternal education level						
College graduate	170	51.4	14.80	(13.91–15.69)	87.23	(84.46–89.969)
Some college	72	21.8	17.96	(15.90–20.02)	97.68	(92.43–102.93)
High school graduate/GED	54	16.3	17.41	(15.42–19.39)	99.64	(93.42–105.86)
Less than high school	35	10.6	18.66	(16.05–21.27) **	101.76	(94.54–108.98) **
Race						
White	151	45.6	15.99	(14.94–17.04)	89.93	(86.99–92.87)
Black	155	46.8	16.65	(15.37–17.93)	95.82	(92.03–99.62)
Hispanic	10	3.0	18.10	(15.75–20.45)	97.50	(87.90–107.10)
Other	15	4.5	15.07	(12.41–17.73)	92.47	(81.96–102.97)
Parity						
Primiparous	125	37.8	16.02	(14.86–17.18)	92.16	(88.79–95.53)
1–3	112	33.8	17.08	(15.60–18.56)	93.62	(89.41–97.82)
≥4	94	28.4	15.81	(14.37–17.25)	93.37	(88.86–97.87)
Maternal ADHD Symptoms						
Negative screen	292	88.2	15.91	(15.11–16.71)	92.04	(89.64–94.44)
Positive screen	39	11.8	19.41	(16.77–22.05) *	100.35	(93.42–107.29) *
Maternal smoking						
No	262	79.2	15.93	(15.12–16.74)	90.97	(88.63 - 93.31)
Yes	69	20.8	17.80	(15.71–19.89)	100.95	(94.77–107.14) **
Prepregnancy BMI (Body Mass Index)						
Underweight	21	6.3	13.90	(11.04–16.77)	79.90	(73.47–86.33)
Normal weight	137	41.4	15.64	(14.53–16.75)	91.58	(88.27–94.88)
Overweight	68	20.5	15.93	(14.15–17.70)	94.59	(89.35–99.83)
Obese class I	38	11.5	15.50	(13.83–17.17)	91.27	(86.28–96.260
Obese class II and greater	67	20.2	19.33	(17.23–21.43) *	99.41	(93.11–105.70) *
Weight gain during pregnancy, IOM (Institute of Medicine) category						
Less than recommended	52	15.7	18.58	(16.40–20.75)	95.33	(89.80–100.87)
Recommended	79	23.9	14.42	(13.05–15.78)	88.07	(83.62–92.53)
More than recommended	200	60.4	16.49	(15.48–17.49) *	94.18	(91.18–97.18)
Maternal Diabetes Status						
Normal	281	84.9	15.96	(15.14–16.78)	92.09	(89.73–94.45)
Gestational Diabetes	23	7.0	21.57	(17.53–25.60)	105.65	(92.51–118.79)
Type I Diabetes	10	3.0	17.50	(14.30–20.70)	99.50	(88.14–110.86)
Type II Diabetes	17	5.1	14.53	(11.68–17.38) *	85.80	(77.62–93.98) *
Gestational age, weeks						
34–36	17	5.1	16.06	(13.49–18.63)	92.13	(84.03–100.22)
37–40	237	71.6	16.86	(15.93–17.79)	94.48	(91.75–97.21)
>41	77	23.3	14.71	(13.10–16.33)	88.53	(83.83–93.24)
Birthweight (g)						
< 2500	27	8.2	16.30	(13.87–18.72)	94.73	(86.94–102.52)
2500–4200	289	87.3	16.51	(15.66–17.35)	93.38	(90.90–95.87)
>4200	15	4.5	12.80	(9.97–15.63)	83.00	(76.08–89.92)
Child gender						
Male	171	51.7	16.83	(15.70–17.96)	92.86	(89.54–96.17)
Female	160	48.3	15.78	(14.72–16.83)	93.16	(90.02–96.31)

ANOVA comparison: * *p* < 0.01, ** *p* < 0.001; BASC ADHD = Sum score of BASC hyperactivity and attention problem; BRIEF GEC = Sum of scale raw score of BRIEF inhibit, shift, control, working memory and plan/organize.

**Table 2 ijerph-16-00667-t002:** Association between pre-pregnancy BMI, gestational weight gain, and childhood BASC ADHD (*n* = 331).

Model	ADHD				Hyperactivity				Attention Problem			
	*B*	*se*	*β*	*p*	*B*	*se*	*β*	*p*	*B*	*se*	*β*	*p*
Model 1												
BMI (continuous)	**0.17**	**0.05**	**0.20**	**0.001**	**0.09**	**0.04**	**0.16**	**0.02**	**0.08**	**0.02**	**0.21**	**<0.001**
Model 2												
BMI (categorical)												
0–24.9 (referent)												
25–29.9	0.04	1.05	0.002	0.97	−0.39	0.82	−0.03	0.64	0.47	0.41	0.06	0.25
30–34.9	−0.44	0.91	-0.02	0.63	−0.64	0.64	−0.04	0.32	0.22	0.52	0.02	0.67
>35	**4.22**	**1.34**	**0.21**	**0.002**	**2.43**	**1.05**	**0.18**	**0.02**	**1.79**	**0.50**	**0.20**	**<0.001**
Model 3												
Gestational weight gain (kg) (continuous)	0.01	0.05	0.02	0.80	−0.02	0.04	−0.02	0.69	0.03	0.02	0.07	0.26
Model 4												
Gestational weight gain (categorical)												
Less than adequate	**2.74**	**1.25**	**0.15**	**0.03**	**2.04**	**0.89**	**0.16**	**0.02**	0.70	0.56	0.09	0.22
Adequate (referent)												
More than adequate	1.16	0.88	0.08	0.19	0.52	0.61	0.05	0.39	0.62	0.37	0.10	0.09

*B*: unstandardized regression coefficient; *se*: robust standard error; *β*: standardized coefficient; p: p-value; Models 1 and 2 adjusted for gestational diabetes, parity, birth weight, child’s age, sex, mother’s age, race, education, gestational weeks, smoking and mother’s concurrent ADHD; Models 3 and 4 adjusted for pre-pregnancy BMI, parity, birth weight, child’s age, sex, mother’s age, race, education, gestational weeks, smoking and mother’s ADHD. Bold format indicated statistically significant association at *p* < 0.5.

**Table 3 ijerph-16-00667-t003:** Association between pre-pregnancy BMI, gestational weight gain, and childhood BRIEF GEC score (*n* = 331).

Model	GEC Score				Inhibit				Shift				Emotional Control				Working Memory				Plan/Organize			
	*B*	*se*	*β*	*p*	*B*	*se*	*β*	*p*	*B*	*se*	*β*	*p*	*B*	*se*	*β*	*p*	*B*	*se*	*β*	*p*	*B*	*se*	*β*	*p*
Model 1																								
BMI (continuous)	**0.38**	**0.16**	**0.16**	**0.02**	**0.09**	**0.04**	**0.13**	**0.04**	**0.06**	**0.03**	**0.16**	**0.02**	0.04	0.03	0.08	0.23	**0.12**	**0.05**	**0.17**	**0.02**	**0.07**	**0.02**	**0.18**	**0.004**
Model 2																								
BMI (categorical)																								
0–24.9 (referent)																								
25–29.9	1.02	2.81	0.02	0.72	0.13	0.92	0.01	0.89	0.16	0.44	0.02	0.72	−0.41	0.49	−0.04	0.41	0.93	0.94	0.06	0.33	0.20	0.50	0.02	0.69
30–34.9	−2.50	3.35	−0.04	0.46	−1.04	0.99	−0.05	0.29	−0.79	0.61	−0.07	0.20	−0.90	0.79	−0.07	0.25	−0.09	0.97	−0.005	0.92	0.20	0.59	0.02	0.73
>35	**9.31**	**4.30**	**0.16**	**0.03**	**2.23**	**1.14**	**0.13**	**0.05**	**1.41**	**0.70**	**0.14**	**0.05**	1.04	0.76	0.09	0.17	**3.21**	**1.35**	**0.19**	**0.02**	**1.42**	**0.71**	**0.14**	**0.05**
Model 3																								
Gestational weight gain (kg) (continuous)	0.27	0.14	0.11	0.06	0.05	0.04	0.07	0.23	0.03	0.02	0.07	0.20	0.03	0.03	0.06	0.29	**0.08**	**0.04**	**0.10**	**0.05**	**0.07**	**0.02**	**0.15**	**0.01**
Model 4																								
Gestational weight gain (kg) (categorical)																								
Less than adequate	1.66	3.83	0.03	0.67	0.80	1.07	0.06	0.45	0.38	0.62	0.04	0.54	0.70	0.73	0.07	0.34	−0.11	1.13	−0.01	0.92	−0.79	0.68	−0.09	0.24
Adequate (referent)																								
More than adequate	2.86	2.85	0.07	0.32	0.66	0.79	0.06	0.41	0.39	0.47	0.06	0.41	0.40	0.53	0.05	0.45	0.41	0.85	0.03	0.63	0.26	0.48	0.04	0.59

*B*: unstandardized regression coefficient; *se*: robust standard error; *β*: standardized coefficient; *p*: *p*-value; Models 1 and 2 adjusted for gestational diabetes, parity, birth weight, child’s age, sex, mother’s age, race, education, gestational weeks, smoking and mother’s concurrent ADHD; Models 3 and 4 adjusted for pre-pregnancy BMI, parity, birth weight, child’s age, sex, mother’s age, race, education, gestational weeks, smoking and mother’s ADHD. Bold format indicated statistically significant association at *p* < 0.5.

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
