# Peer review of "Pre-Pregnancy Weight and Symptoms of Attention Deficit Hyperactivity Disorder and Executive Functioning Behaviors in Preschool Children"

_ijerph, 2019, doi:10.3390/ijerph16040667_

Round 1
Reviewer 1 Report
That you for the opportunity to review this very interesting paper that examines the association among pre-pregnancy BMI, gestational weight gain and symptoms of ADHD and executive function.
This is a well written paper with sound methodological analysis however, I do have a few concerns that should be addressed prior to publication.
1) It seems as if the wrong range of weight gain is being used for the obese category. The 2009 IOM range is 5-9 kg, thus it is not clear where 6kg comes from?
2) There is huge sample attrition for the outcomes measured and this affects the internal validity of the results. The authors state that 1272 moms of children between the ages of 2 and 6 were asked to complete the outcome assessments but only 1/3 did so. While the authors provide comparisons for the full sample (not the 1272) to the final sample it is clear that the comparison should be to the 1272 and some sort of statistical method should be used to correct for the bias in who responded. Perhaps using a two part model that is run simultaneously where the first model predicts who responded and then the second model is the association of interest? In any regard, more attention to this attrition and how it might influence the results is needed in both the methods and limitations sections.
3) While the authors conducted a sub-study to check on the validity of the self-reported weights, which is great, and found high correlation what we don't know is how does this influence their calculation of GWG? I am also a bit confused because on page 3 line 117, it states that GWG was computed using the nurses measured weight and the last weight prior to delivery or at delivery. What was the gestational age of this measured weight? How was this taken into account when categorizing women into adequacy of weight gain?
4) Lastly, somewhat connect to the second point. The sample size for the analysis is roughly 300 women with approximately 11 to 16% prevalence for the outcomes measured which indicates to me that they had very low power to detect the differences noted with the exception of when a continuous form of the variable was used. This needs to be addressed a bit more.
Author Response
Dear Editors,
We are pleased to re-submit for publication the revised version of “Pre-pregnancy weight and symptoms of attention deficit hyperactivity disorder and executive functioning behaviors in preschool children” to the International Journal of Environmental Research and Public Health. We are appreciative of the time and effort provided by the two reviewers who have provided us with constructive criticisms and suggested edits for improvement. We have worked to address each concern, as described and outlined below. Please note that reviewer comments are italicized, while responses are in regular type font.
____________________________________________________________________________________
REVIEWER 1
Thank you for the opportunity to review this very interesting paper that examines the association among pre-pregnancy BMI, gestational weight gain and symptoms of ADHD and executive function. This is a well-written paper with sound methodological analysis however, I do have a few concerns that should be addressed prior to publication.
1) It seems as if the wrong range of weight gain is being used for the obese category. The 2009 IOM range is 5-9 kg, thus it is not clear where 6kg comes from?
Reply: The 2009 IOM ranges were used while performing analysis. Yet, as Reviewer 1 has pointed out, the description in the methods needed to be corrected to reflect these range recommendations. We have addressed this concern by changing the sentence to read,
“Categories for gestational weight gain were constructed based on the Institute of Medicine (IOM) guidelines of 12.5-18 kg weight gain for women with pre-pregnancy weight of <19.8 kg/m2; 11.5-16 kg weight gain for women 19.8-26.0 BMI; 7-11.5 kg/m2 for women 26-29 kg/m2 and 5-9 kg for women >= 30 kg/m2 [27]” (as described on page 3).
2) There is huge sample attrition for the outcomes measured and this affects the internal validity of the results. The authors state that 1272 moms of children between the ages of 2 and 6 were asked to complete the outcome assessments but only 1/3 did so. While the authors provide comparisons for the full sample (not the 1272) to the final sample it is clear that the comparison should be to the 1272 and some sort of statistical method should be used to correct for the bias in who responded. Perhaps using a two part model that is run simultaneously where the first model predicts who responded and then the second model is the association of interest? In any regard, more attention to this attrition and how it might influence the results is needed in both the methods and limitations sections.
Reply: This point was also raised by Reviewer 2 below. We can see that there was some confusion about some of these numbers and we have clarified this in the text (page 3, line 96) stating the following to the manuscript:
“Between January 2011 and June 2013, mothers from the NEST study, who had children between the ages of 2 and 6 years of age (n=1278) were sent a survey by research staff, that included measures of ADHD symptoms and related executive behaviors. Of those who were sent a survey, 223 were undeliverable and 165 were deemed ineligible due to the child’s age at the time the survey was completed. We received 531 surveys (531/ (1272-165) = 48.0% response rate). Of the 531 returned surveys, 331 had data on all data needed for this study. Main sources of missing data were due to one measure (maternal ADHD symptoms) that was added subsequent to sending and initial set of surveys and missing data on pre-pregnancy weight and data on gestational diabetes from the medical records.”
As suggested we now compare the 331 in our analysis sample to the sample who were originally sent surveys minus the 331 in our analysis sample and the 165 who were deemed ineligible. Please refer to page 3, line 108 for details on these comparisons.
3) While the authors conducted a sub-study to check on the validity of the self-reported weights, which is great, and found high correlation what we don't know is how does this influence their calculation of GWG? I am also a bit confused because on page 3 line 117, it states that GWG was computed using the nurses measured weight and the last weight prior to delivery or at delivery. What was the gestational age of this measured weight? How was this taken into account when categorizing women into adequacy of weight gain?
Reply: We do not believe that our use of self-report pre-pregnancy weight at the last menstrual period and nurse-measured weight at delivery should be biased in any way since the self-report assessment of weight corresponds with the measured assessment of weight. The statement referred (page 3, line 133) has been changed to read:
“Gestational weight gain was computed as the difference between the weight of the last menstrual period (prior to the first prenatal visit), and the last weight taken up to seven days before or at delivery from medical records.”
We did not take into account gestational age in the calculation of GWG, however, our design for our analyses did take gestational age into account.
4) Lastly, somewhat connect to the second point. The sample size for the analysis is roughly 300 women with approximately 11 to 16% prevalence for the outcomes measured which indicates to me that they had very low power to detect the differences noted with the exception of when a continuous form of the variable was used. This needs to be addressed a bit more.
Reply: Indeed, this is correct. The percentage of children with BASC HY or AP scores in the clinical significant rages in our sample was approximately close to what one would expect in the population (expected ranges for BASC HY and AP are about 14-17%), but the small sample in our study results in a lower absolute number. For this reason and because we were interested in assessing the dimensional construct we did not dichotomize our data on the outcome variables. In this way, we conserve statistical power and are able to assess the hypothesized relationships.

Reviewer 2 Report
This is an observational study examining the longitudinal association between pre-pregnancy body mass index (BMI) and gestational weight gain (GWG) and ADHD symptoms and executive functioning in preschool children. In 331 mother-child pairs (please specify the location), the authors reported associations between pre-pregnancy BMI (>35) and children’s ADHD symptoms, inhibitory control and attention. Inadequate GWG was associated with hyperactive impulsive symptoms while more than adequate GWG was longitudinally associated with attention problems in 2-6 year old children. The findings are very interesting and consistent with several earlier studies in older children. However, major limitations reduce my enthusiasm about the paper.
Substantial loss to follow up is a major limitation of the study. From initial ~2500 participants, about 1300 were invited to participate in the preschool age follow up. Was this group a random selection of initial recruitment? From this number only 37% responded to the surveys and again one third provided incomplete data. With the substantial loss to follow and differential non-response (as reflected in the comparison of 331 included and 2595 initial recruitment), analysis with IPW should be the main approach rather than sensitivity analysis presented in the supplementary materials. Another useful approach is to use MI to impute missing items on BASC, BRIEF-P, or covariates in 468 participants.
Another concern is model selection which included several variables which could potentially be the intermediate factors, for example birth weight and gestational age. Similarly, exclusion of children who were born preterm introduces bias (conditioning on the intermediate variable), because several studies and systematic reviews have shown association between maternal obesity and preterm delivery.
A significant limitation of the study is the use of only parent questionnaire for assessment of both ADHD symptoms and executive functioning. While parents can provide valuable information on their children’s behavior in a naturalistic setting, their report, especially on ADHD symptoms and at a young age remains limited because parents may be affected by many factors in reporting child’s behavior.
Between 11% and 16% of children had behavior and EF T scores above the clinical significant range. Please comment on these % and compare with norm samples from the source population.
Limitations of using medical record to obtain information on maternal weight should also be discussed.
The abstract needs to clarify the gap in the knowledge which justifies the rational for further investigation on maternal obesity and child behavior and EF. Also the location and year/month of recruitment should be presented in the abstract.
Several recent publications on the topic are missed in the reference list of the Introduction, e.g., Windham et al. Autism Research 2018, Girchenko et al. Int J Obesity 2018, Yeung et al. Pediatrics 2017. Also please see Gardner et al. Int J Epi 2015 which used a sibling design and found null association between maternal BMI and ASD. The paper would improve greatly if the authors were more critical in their review of previous publication on this topic and possible underlying mechanisms.
For methods, please follow the STROBE guideline and present information on the setting and location, in addition to relevant dates. What was the range for gestational age at recruitment (instead of SD, please provide range).
Author Response
Dear Editors,
We are pleased to re-submit for publication the revised version of “Pre-pregnancy weight and symptoms of attention deficit hyperactivity disorder and executive functioning behaviors in preschool children” to the International Journal of Environmental Research and Public Health. We are appreciative of the time and effort provided by the two reviewers who have provided us with constructive criticisms and suggested edits for improvement. We have worked to address each concern, as described and outlined below. Please note that reviewer comments are italicized, while responses are in regular type font.
REVIEWER 2
This is an observational study examining the longitudinal association between pre-pregnancy body mass index (BMI) and gestational weight gain (GWG) and ADHD symptoms and executive functioning in preschool children. In 331 mother-child pairs (please specify the location), the authors reported associations between pre-pregnancy BMI (>35) and children’s ADHD symptoms, inhibitory control and attention. Inadequate GWG was associated with hyperactive impulsive symptoms while more than adequate GWG was longitudinally associated with attention problems in 2-6 year old children. The findings are very interesting and consistent with several earlier studies in older children. However, major limitations reduce my enthusiasm about the paper.
1) Substantial loss to follow up is a major limitation of the study. From initial ~2500 participants, about 1300 were invited to participate in the preschool age follow up. Was this group a random selection of initial recruitment? From this number only 37% responded to the surveys and again one third provided incomplete data. With the substantial loss to follow and differential non-response (as reflected in the comparison of 331 included and 2595 initial recruitment), analysis with inverse probability weighting (IPW) should be the main approach rather than sensitivity analysis presented in the supplementary materials. Another useful approach is to use Multiple Imputation to impute missing items on BASC, BRIEF-P, or covariates in 468 participants.
Reply: Regarding our sample and loss to follow-up, please see our response above for Reviewer 1. We did not choose to use multiple imputation for the BASC because the scoring manual suggests that if 2 or more items were missing on a subscale it should not be scored. This, however, only represented a small number (n=18). The main source of missing data where from the Adult ADHD self-report scale (ASRS; n=55). In part, the missingness of this data is because one of our measures of ADHD symptoms was added subsequent to the original set of surveys that were sent. The other large sources of missing data were due to missing data in the medical records on pre-pregnancy BMI (n=23) or gestational diabetes (n=22). We felt that there was a large amount of missing data and the multiple imputation was not appropriate in this case. We appreciate the suggestion to make our IPW the main approach rather than the secondary one, however, we think the traditional method will be more accessible to most readers. The IPW results help to substantiate those using the more traditional method and thus seemed appropriate to include as supplemental.
2) Another concern is model selection which included several variables which could potentially be the intermediate factors, for example birth weight and gestational age. Similarly, exclusion of children who were born preterm introduces bias (conditioning on the intermediate variable), because several studies and systematic reviews have shown association between maternal obesity and preterm delivery.
Reply: We re-ran the models as suggested excluding birth weight and gestational age and dropping the conditioning on gestational age. The results are similar with no changes in significance of the association nor do they change the point estimate by more than 10% suggesting they are not acting as mediators. We have added this to the paper (page 4, line 186). While in general agree with the reviewer we do not want to over fit the model, we retained our original variables for two reasons. First, the choice of covariates was guided by what has been used in previous research summarized by Sanchez et al. (2017) and many papers in this field control for a similar set of putative core covariates. Thus, our findings can be compared with those of others. Second, there may be other factors besides maternal obesity that impact birth weight and gestational age and thus by controlling for them statistically or through exclusion our results represent the relationship between maternal obesity and child outcomes independent of birth weight or gestational age. If the reviewer remains concerned about this choice, we will yield our preference and present the analyses unadjusted for these factors in a subsequent revision.
3) A significant limitation of the study is the use of only parent questionnaire for assessment of both ADHD symptoms and executive functioning. While parents can provide valuable information on their children’s behavior in a naturalistic setting, their report, especially on ADHD symptoms and at a young age remains limited because parents may be affected by many factors in reporting child’s behavior.
Reply: Although the parent-report BRIEF and BASC are widely used and standardized for measuring ADHD symptoms and child executive functioning behaviors, the reviewer is correct to point out that the reliance only on parent-report measures is a weakness of the study. This is explained in the strengths and limitations section of the manuscript, where we have addressed the limitations of using parent-report measures, while emphasizing the need to use individually administered and performance-based measures. Specifically, we state on page 11:
“Additionally, we only have parent questionnaires available for assessment of both ADHD symptoms and executive functioning. Though parents can provide information on their children’s behavior in a naturalistic setting, their report may be limited since parents may be biased in terms of how they report on their children’s behavior. Nevertheless, we did use standardized measures of ADHD and executive behaviors commonly used to assess ADHD. To overcome these measurement challenges, it would be important in future studies to use longitudinal designs with individually administered and performance-based measures (e.g., NIH toolbox, measures of IQ). Having multiple informants, such as the child’s pre-school teachers, would also be warranted in futures studies.”
4) Between 11% and 16% of children had behavior and EF T scores above the clinical significant range. Please comment on these % and compare with norm samples from the source population.
Reply: The percentage of children in our sample was fairly similar to the expected rates. Specifically, norm sample from the BASC-II population presented a clinical significant range of 14-15% for the BASC HY and 14-17% for the BASC AP among children ages 2-6 years. Meanwhile, norm sample from BRIEF population showed a range of 8-10% with abnormal elevation in GEC T-score for boys and girls at the same age group. This information was added to the paper (page 5, line 207) .
5) Limitations of using medical record to obtain information on maternal weight should also be discussed.
Reply: We do not think that the use of medical records to obtain women’s weight is a meaningful limitation. Our ability to have this information so close to the child’s birth is a strength and only possible through access to medical records. Indeed, this is common practice in studies that include gestational weight because it is not feasible to have women weighed by a research staff at their delivery. We feel confident that the weights in the medical records are accurate since they are often used to guide clinical care. Also, since all women were from the same provider network the same scales were used across the network.
6) The abstract needs to clarify the gap in the knowledge which justifies the rational for further investigation on maternal obesity and child behavior and EF. Also the location and year/month of recruitment should be presented in the abstract.
Reply: Changes are made in the abstract as Reviewer 2 suggested.
7) Several recent publications on the topic are missed in the reference list of the Introduction, e.g., Windham et al. Autism Research 2018, Girchenko et al. Int J Obesity 2018, Yeung et al. Pediatrics 2017. Also please see Gardner et al. Int J Epi 2015 which used a sibling design and found null association between maternal BMI and ASD. The paper would improve greatly if the authors were more critical in their review of previous publication on this topic and possible underlying mechanisms.
Reply: We have included the recent publications that as Reviewer 2 suggested. We also included descriptions in introduction to state the overall trends in the findings across these studies as well as null results.
“Of note, however, studies have also documented null associations between pre-pregnancy weight and neurodevelopmental outcomes [16-17], and discordant sibling design studies suggested that the association may not be causal [18-19]. Given some of the differences in findings, continued research using different samples and different methods to better understand this association between pre-pregnancy obesity and child neurodevelopmental outcomes.”
8) For methods, please follow the STROBE guideline and present information on the setting and location, in addition to relevant dates. What was the range for gestational age at recruitment (instead of SD, please provide range).
Reply: We followed the STROBE guideline to present information on initial cohort NEST. To clarify, information on the setting and location for the follow-on study was also described, as Reviewer 2 suggested. Additionally, we have provided the range for gestational age at recruitment.

Round 2
Reviewer 2 Report
Thanks for clarifying about your sample selection. However, the response rate of 48% is still concerning and, considering that, the correct approach remains to be the models using IPW. The goal is to perform analysis in an unbiased way; whether the readers are familiar with the epidemiological method or not shouldn’t define the methodology of a study. Similarly, because “many papers in this field control for a similar set of putative core covariates” cannot be a good reason for covariate selection. Please consider including the most parsimonious model in the manuscript.
Author Response
Dear reviewer,
As advised, we have revised our method for presenting the results. In the current revision, we use the IPW method as the main approach, rather than presenting it as supplementary. To do this we went back and recalculated our models using the inverse probability method described by Rinsky et al., Am. Journal of Epidemiology, 2017, which accounts for non-response and covariate imbalance on the exposure (i.e., BMI or gestational weight gain). The findings remain largely the same with very little changes to the point estimates and p-values. We thank the reviewer for their feedback and hope that the reviewer will find the new version acceptable. We believe, and hope the reviewer agrees, that the revised version is much stronger.
Round 3
Reviewer 2 Report
I have no further comment.